# Energy Dissipation in the Human Red Cell Membrane

**DOI:** 10.3390/biom13010130

**Published:** 2023-01-09

**Authors:** Thomas M. Fischer

**Affiliations:** Laboratory for Red Cell Rheology, Krummer Weg 20, 52134 Herzogenrath, Germany; thmfischer@gmail.com; Tel.: +49-160-2293318

**Keywords:** redistribution of the membrane skeleton, lateral drag of the Band 3 complexes through the lipids of the bilayer, single layer bending, bilayer couple bending, local bending, global bending, lateral drag of the Band 3 complexes through the lipids of each monolayer separately

## Abstract

The membrane of the human red cell consists of a lipid bilayer and a so-called membrane skeleton attached on the cytoplasmic side of the bilayer. Upon the deformation of red cells, energy is dissipated in their cytoplasm and their membrane. As to the membrane, three contributions can be distinguished: (i) A two-dimensional shear deformation with the membrane viscosity as the frictional parameter; (ii) A motion of the membrane skeleton relative to the bilayer; (iii) A relative motion of the two monolayers of the bilayer. The frictional parameter in contributions (ii) and (iii) is a frictional coefficient specific for the respective contribution. This perspective describes the history up to recent advances in the knowledge of these contributions. It reviews the mechanisms of energy dissipation on a molecular scale and suggests new ones, particularly for the first contribution. It proposes a parametric fitting expected to shed light on the discrepant values found for the membrane viscosity by different experimental approaches. It proposes strategies that could allow the determination of the frictional coefficients pertaining to the second and the third contribution. It highlights the consequences characteristic times have on the state of the red cell membrane in circulation as well as on the adaptation of computer models to the red cell history in an in vitro experiment.

## 1. Introduction

Compared to other cells in the human body, the structure of red cells is relatively simple. A closed membrane is filled with a concentrated solution of hemoglobin—from a mechanical point of view a Newtonian liquid. The shape is discoidal with average values of diameter and thickness of 8 µm and 2 µm, respectively. The membrane consists of a lipid bilayer and a so-called membrane skeleton attached to the cytoplasmic side of the bilayer. The lipid bilayer is referred to as a bilayer in the following. From a mechanical point of view, it can be considered a two-dimensional (2D) liquid. It provides bending stiffness to the membrane. Under the experimental situations considered here, the bilayer is essentially incompressible. So far, it also conserves the surface area of the membrane. Embedded in the bilayer are so-called intrinsic proteins spanning the bilayer. The membrane skeleton is referred to as the skeleton in the following. Its main constituents are the proteins spectrin and actin forming a 2D (loosely knitted) lattice. The skeleton provides shear elasticity to the membrane. It is connected to the intrinsic proteins Band 3 and glycophorin.

The coefficients describing the mechanical behavior of the membrane have been measured by comparison between experiments and adapted theoretical descriptions. Among these coefficients is the 2D viscosity of the red cell membrane. By comparison with the orders of magnitude lower value of the bilayer, the respective energy dissipation was considered to occur in the skeleton. Although the determination of the membrane viscosity became standard in the characterization of membrane mechanical properties, an attempt to explain the responsible energy dissipation on a molecular scale was never made.

With the advances in computational technology, membrane models of increasing sophistication were developed. (These models typically treat the membrane as a 2D sheet with no thickness. This has usually been justified by comparing red cell dimensions with the bilayer thickness of ~4 nm. By lowering a force-sensitive probe onto the membrane, Heinrich et al. [1] observed the first elastic recoil at a skeleton thickness of ~90 nm. Nevertheless, the 2D idealization is commonly adopted in modeling.) Among these, four recent studies [2,3,4,5] presented parametric fits of the membrane viscosity to existing experimental results. However, the results of these studies suggest that additional sources of energy dissipation have to be considered to describe the dynamic behavior of the red cell.

The paper is organized as follows. Section 2 reviews the collection of the knowledge with respect to the energy dissipation in the membrane by (i) Dynamic membrane shear with the membrane viscosity as the frictional parameter; (ii) Skeleton slip; i.e., the relative motion of bilayer and skeleton; (iii) Monolayer slip; i.e., the relative motion of the two monolayers of the bilayer. The frictional parameter in contributions (ii) and (iii) is a frictional coefficient specific to the respective contribution. Section 3 presents possible mechanisms on a molecular scale and illuminates some of them by reference to particular experiments. Section 4 proposes a parametric fitting expected to shed light on the discrepant values found for the membrane viscosity by different experimental approaches. Furthermore, strategies are proposed that could allow the determination of the friction coefficients pertaining to the second and the third contribution. Section 5 highlights the consequences the various characteristic times have on the state of the red cell membrane in the circulation as well as on the adaptation of computer models to the red cell history in an in vitro experiment. Section 6 ties the presented information together and ventures an outlook.

## 2. History of the Consideration of the Energy Dissipation in the Red Cell Membrane

Section 2.1 deals with the various methods used to determine the membrane viscosity. A matter of concern were and, to some extent, still are the different values obtained by different methods thus shedding doubt on the membrane viscosity being the sole contribution to the energy dissipation in the red cell membrane. The findings of Rezghi and Zhang [5] added to this suspicion. Section 2.2 and Section 2.3 describe the mechanisms of the two additional contributions to energy dissipation. No parametric fits were undertaken for these two cases. For the second contribution, theoretical estimates of the involved frictional coefficient differ by three orders of magnitude thus demonstrating the necessity of a direct determination by parametric fits to appropriate experiments.

### 2.1. Dynamic Membrane Shear—The First Contribution to the Energy Dissipation

The shear stiffness was evaluated by aspirating a membrane tongue into small bore glass pipettes and the shear modulus (μ) was determined as 7 µN/m [6]. In those days (1970s–1990s), the skeleton was considered to be incompressible—equivalent to local conservation of its surface area. The first account of the viscosity ($\bar{\eta_{\mathrm{mem}}}$, where the overline indicates that a single value was determined) used the time constant of relaxation of a membrane tongue after an aspirated cell was expelled from the pipette [7]. $\bar{\eta_{\mathrm{mem}}}=10^{-6}$ Pa s m was given as an order of magnitude estimate. Since this value was three orders of magnitude above what was estimated to be the viscosity of the bilayer, $\bar{\eta_{\mathrm{mem}}}$ was considered to be the viscosity of the skeleton.

Chien et al. [8] observed the time courses during the aspiration of a membrane tongue into small bore pipettes and during its release. Several characteristic times were recorded, $\tau_{1}$ and $\tau_{2}$ during the entry of the tongue and $\tau_{3}$ upon release of the aspiration pressure. The shear modulus was measured as μ=4.2 µN/m and the membrane viscosity was calculated according to $\bar{\eta_{\mathrm{mem}}}=\mu \tau$. From $\tau_{1}$, $\bar{\eta_{\mathrm{mem}}}$ was found in the range of (0.6–4) $\cdot \;10^{-7}$ Pa s m varying inversely with the initial rate of entry which in turn depended on the final tongue length. Based on the $\tau_{2}$ results, the membrane viscosity was $2\cdot 10^{-5}$ Pa s m. Using $\tau_{3}$, the membrane viscosity was estimated as $5.44\cdot 10^{-7}$ Pa s m after 20 s of deformation. Upon reduction of this time interval down to 2 s, $\tau_{3}$ decreased to approach the low values of $\tau_{1}$. As an explanation for the complicated behavior, the authors suggested a shear thinning lasting over a certain period of time. Tözeren et al. [9] interpreted the two time scales upon entry as a creep phenomenon and presented a description in closed form. Fischer [10] ascribed $\tau_{2}$ to skeleton slip introduced in Section 2.2.

Hochmuth et al. [11] observed the relaxation of a red cell after being released from a static extension at two diametrically opposed locations on the membrane. From the characteristic time of the relaxation (τ=0.1−0.13 s) and μ=6 µN/m, $\bar{\eta_{\mathrm{mem}}}=$ 6–8 $\cdot 10^{-7}$ Pa s m was determined. Without going into detail, it is mentioned that this concept was later used in numerous studies by stretching the red cells using pipettes, optical tweezers, electrical fields or shear flows.

Tran-Son-Tay et al. [12] considered red cells suspended in a solution of viscosity ($\eta_{0}$) of 35 mPa s and subjected to simple shear flow with shear rate ($\dot{\gamma }$). In a steady state, the red cells display an elongated shape and a continuous flow of the membrane around that shape [13]. This particular motion of the membrane has been called the tank-tread (TT) motion [14]. The kinematics of the TT motion are sketched in Figure 1. Tran-Son-Tay et al. [12] collected the geometric data of the red cells and their TT frequency (TTF). Equating the power supplied to the red cell from the shear flow with the energy dissipated in the red cell, they deduced $\bar{\eta_{\mathrm{mem}}}$. The value depended moderately on $\dot{\gamma }$. At $\dot{\gamma }=29$/s they found $\bar{\eta_{\mathrm{mem}}}=1.03\cdot 10^{-7}$ Pa s m and at $\dot{\gamma }=171$/s the value was $0.86\cdot 10^{-7}$ Pa s m, an order of magnitude below the value given by Hochmuth et al. [11].

Later this group refined the treatment and determined $\eta_{\mathrm{mem}}$ depending on the location on the red cell membrane [16]. The range of these values even overlapped with that found for pure lipid bilayers [17].

With the appearance of discretized computer models, previous determinations of $\bar{\eta_{\mathrm{mem}}}$ were reevaluated by fitting these models to experimentally observed TTFs [18]. As in Tran-Son-Tay et al. [12], a decrease of $\bar{\eta_{\mathrm{mem}}}$ with the increase in $\dot{\gamma }$ was found. Tsubota [3] fitted data collected at $\eta_{0}=54$ mPa s and $\dot{\gamma }$ from 8/s to 260/s and found $\bar{\eta_{\mathrm{mem}}}$ to decrease from 4 to 1.5 $\cdot 10^{-7}$ Pa s m. Matteoli et al. [2] used data at $\eta_{0}=13$ mPa s from $\dot{\gamma }=30$/s to 260/s and at $\eta_{0}=54$ mPa s from $\dot{\gamma }=8$/s to 130/s and they found a common decrease of $\bar{\eta_{\mathrm{mem}}}$ from 24 to 2.8$\cdot 10^{-8}$ Pa s m. It is surprising that the values of $\bar{\eta_{\mathrm{mem}}}$ at the same values of $\dot{\gamma }$ and $\eta_{0}$ are quite different in these two studies, although the axis ratios of the respective stress-free shapes were similar (Table 1).

The determination of $\bar{\eta_{\mathrm{mem}}}$ via the recovery time was revisited by Guglietta et al. [4]. Comparison of their parametric study with the relaxation data of Mills et al. [19] results in $\bar{\eta_{\mathrm{mem}}}\approx 4.2\cdot 10^{-7}$ Pa s m. The difference to the respective values determined via TTing is smaller (line 4 and 5 in Table 2) than before (line 2 and 3 in Table 2) but is still existing.

Rezghi and Zhang [5] criticized the studies by Matteoli et al. [2] and Tsubota [3] in that the same $\eta_{\mathrm{mem}}$ value was used for all membrane elements instead of determining $\eta_{\mathrm{mem}}$ as a function of the local 2D membrane shear rate as one would expect for a non-Newtonian viscosity. Actually, this consideration had been adopted by Sutera et al. [16], where the membrane viscosity was assumed as a function of the second invariant of the surface strain rate. Rezghi and Zhang [5] fitted to TTFs collected at four values of $\eta_{0}$ (13, 29, 54, and 109 mPa s). In contrast to the authors in Table 1, they used the biconcave disc as the stress-free shape. A multitude of values for $\bar{\eta_{\mathrm{mem}}}$ was reported which for brevity are not repeated here. In agreement with the authors in Table 1, the trend in these data indicates a decrease of $\bar{\eta_{\mathrm{mem}}}$ with $\dot{\gamma }$. On the other hand, comparison with experimental data on the angle of inclination during TTing [20] yielded a constant value for $\bar{\eta_{\mathrm{mem}}}$. Furthermore, fitting the elongation of TTing red cells [21], even resulted in an increase of $\bar{\eta_{\mathrm{mem}}}$ with $\dot{\gamma }$. Among other explanations, the authors conjectured that the fact that different blood samples had been used in the measurement of these three parameters might be responsible for the different behavior. However, such drastic discrepancies call for a widened view and to include additional sources of energy dissipation in the theoretical models.

### 2.2. Skeleton Slip—The Second Contribution to the Energy Dissipation

Let us go back to the early days when the skeleton was considered to be incompressible. Stokke et al. [22], considering the skeleton as an ionic gel, widened the view by claiming that the skeleton has finite compressibility. This means that besides the shear modulus a biaxial modulus describing the resistance of the skeleton to a change in the area has to be considered. If the red cell shape differs from the reference configuration of the skeleton, the minimum in both its shear and biaxial strain energy requires its rearrangement towards a non-uniform density on the membrane [23]. Provided the biaxial modulus is not much larger than the shear modulus, a non-uniform skeleton density becomes apparent in deformations with strong non-uniform local shear strains. An example is the aspiration of a membrane tongue into a small bore glass pipette. Fischer [10] predicted that the skeleton density would increase at the pipette tip and decrease at the tip of the aspirated membrane tongue. This redistribution was demonstrated experimentally by Discher et al. [24].

A dynamic redistribution gives rise to a second source of energy dissipation within the membrane. In mechanical terms, a redistribution is equivalent to a slip between the skeleton and the bilayer. Accordingly, its contribution to energy dissipation is called skeleton slip in the following. The mechanical effect of the skeleton slip can be represented by:(1)$$\sigma_{\mathrm{sb}}=c_{f}\cdot \Delta v_{\mathrm{sb}}$$
where $\sigma_{\mathrm{sb}}$ is shear stress opposing the slip between skeleton and bilayer, $c_{f}$ is the respective frictional coefficient, and $\Delta v_{\mathrm{sb}}$ is the difference in velocity between the skeleton and the bilayer.

Since the skeleton is connected to the bilayer via intrinsic proteins, skeleton slip involves a relative motion of these proteins and the bilayer. A first approach to quantify the lateral drag of these anchor proteins through the bilayer was made by Fischer [10]. The drag was estimated using a theoretical expression of the force necessary to move a circular cylinder embedded in a thin liquid sheet of finite size [25].

Peng et al. [26] estimated $c_{f}$ using the lateral diffusivity of the intrinsic anchor protein Band 3. They determined $c_{f}=\left(\rho /\rho_{0}\right)144\cdot 10^{6}$ Ns/m³ where ρ is the actual and $\rho_{0}$ the undisturbed density of this protein. Accounting for both membrane shear and skeleton slip, Peng and Zhu [27] modeled TTing red cells. The viscosities of cytoplasm and suspending phase were both set at 13 mPa s, and the shear rate $\dot{\gamma }=$ 200/s and $\bar{\eta_{\mathrm{mem}}}=$ 5 $\cdot 10^{-8}$ Pa s m [12] were used. The results showed that it took about one minute or 392 TT cycles to reach the steady state in the distribution of skeleton density. As to the cell elongation, the initial fast rise was followed by a small decrease in parallel to the attainment of the steady state in skeleton distribution. The TTF remained basically constant.

Accounting for dynamic membrane shear alone, the steady state in red cell elongation is attained within a few TT cycles [26]. Accordingly, the initial fast rise in red cell elongation in [26] can be considered to be controlled by dynamic membrane shear. The slight decrease in red cell elongation can be attributed to a small decrease in energy dissipation after the redistribution of the skeleton has reached a steady state.

In their determination of $c_{f}$, Peng et al. [26] used the larger of the two values determined for the diffusion coefficient by Kodippili et al. [28]. As discussed by the last authors, the attribution of the two values to the different fractions of Band 3 is presently unclear. Had Peng et al. [26] used the lower value, the redistribution would have been calculated to occur an order of magnitude more slowly. On the other hand, the use of the experimental diffusion coefficients has been criticized by Turlier et al. [29]. Based on an equation of Saffman and Delbrück [30], these authors calculated a value for the diffusion coefficient of free Band 3 two orders of magnitude larger (Their estimate was based on the diameter of the naked Band 3 molecule. After correction to the average diameter of the Band 3 complexes the ratio decreased somewhat but the two orders of magnitude remained.) than the value used by Peng et al. [26]. This discrepancy calls for an experimental determination of $c_{f}$.

### 2.3. Monolayer Slip—The Third Contribution to the Energy Dissipation

In the early days of red cell mechanics studies, the bending stiffness of the red cell membrane was considered to result from the low compressibility of the two monolayers of the bilayer and their fixed distance [31]. Assuming that the biaxial modulus (*K*) of the bilayer is distributed equally between the two monolayers, the respective bending stiffness ($B_{\mathrm{bc}}$) reads [31]:(2)$$B_{\mathrm{bc}}=\frac{Kh^{2}}{4}$$
where *h* is the distance between the neutral planes of the two monolayers. The motivation for the index “bc” will become apparent below.

Helfrich [32] on the other hand, by analogy to liquid crystals, claimed an intrinsic bending stiffness of each monolayer. The sum of the intrinsic bending stiffnesses of both monolayers would constitute the intrinsic bending stiffness of the bilayer. In the beginning, this stiffness was deduced from the observation of membrane flickering of phospholipid vesicles and tubes, e.g., [33,34]. Later Waugh and coworkers used pulling of so-called tethers of the bilayer to this end [35,36].

To distinguish these two mechanisms, the intrinsic bending stiffness is referred to as single-layer bending (slb) hereafter, and the mechanism described by Equation (Equation 2) is referred to as bilayer couple bending (bcb). In the literature, slb is often referred to as local bending because its elastic energy depends on the local curvature provided the lateral distribution of the different lipid species is uniform on the membrane. Bilayer couple bending is often referred to as global bending because in mechanical equilibrium its elastic energy depends on the average of all curvatures on the membrane. Here comes the third contribution to the energy dissipation into play.

If red cell deformations induce fast changes of local curvature, inner and outer monolayers are stressed in the opposite sense—in the above-introduced nomenclature: local bcb. At locations with positive curvatures, the outer layer is extended and the inner one compressed. As to the sign convention, a spherical membrane is defined to have a positive curvature. In a closed membrane, a positive change in curvature at some location requires a negative change at another location. This leads to a difference in lateral pressure in both monolayers which is relieved by a flow of their lipids in opposite directions—local bcb relaxes to global bcb. In the above-mentioned experiments on flickering [33,34], the changes in curvature are small. Obviously, bcb was essentially global during the experiments and the effects of slb were detected only.

The flow in opposite directions of the lipids in the two monolayers requires their flow relative to the intrinsic proteins which can be considered stationary if we conceptually assume that bending and relaxation occur consecutively. The respective frictional contribution was estimated [37] using calculations of the flow through regular arrays of parallel solid cylinders [38]. It was found that the influence of local bcb during tether extension [39] and during TTing $\gamma_{\mathrm{rheo}}=42$ /s, in ref [40] cannot be neglected.

Besides the differential flow around the intrinsic proteins, the friction involved in the opposite motion of the interdigitating ends of the hydrocarbon chains of the inner and outer monolayers contributes as well. The respective frictional coefficient was estimated using two types of vesicle [41]. One consisted of a pure phospholipid. The other one was a binary mixture. A single order of magnitude was given for the frictional coefficient ($10^{8}\;\mathrm{Ns}/m^{3}$).

Using a coarse-grained molecular dynamics simulation of a bilayer with embedded proteins, Khoshnood et al. [42] found an order of magnitude value for the frictional coefficient of $10^{6}\;\mathrm{Ns}/m^{3}$. The boundary condition was shear stress acting on both sides of the simulated bilayer which is of course different from that driving monolayer slip. Furthermore, as mentioned by the authors, coarse-grained molecular dynamics simulations typically underestimate the actual values.

## 3. Mechanisms on a Molecular Scale

Historically, the origin of $\bar{\eta_{\mathrm{mem}}}$ was attributed to the skeleton proper and that solely by exclusion. Although its value was determined in numerous studies for almost half a century, a mechanism for dynamic membrane shear on a molecular scale has not been suggested so far. This gap is filled in Section 3.1. Most mechanisms for the second and third contributions have been described in Section 2. For the second contribution, an additional mechanism is put forward which, however, depends on the boundary conditions of the respective experiment (Section 3.2).

### 3.1. Dynamic Membrane Shear

Here are several mechanisms that could be responsible for the membrane shear viscosity:**1a** Stretching single spectrin monomers resulted in a higher tension when the stretching rate was increased from 0.08 to 0.8 µm/s [43]. This indicates a viscous resistance against stretch. To set these rates in perspective to experiments, red cell TTing is considered. Using the images shown in [40], the stretch of a single spectrin dimer oriented at a right angle to the flow direction is estimated based on the difference of half the perimeter minus the width of the TTing red cell. Multiplication with 4·TTF gives 0.2, 0.7, 1.5, and 4 µm/s at $\dot{\gamma }=$ 42, 125, 238, and 575/s, respectively.**1b** It is conceivable that the flow of the cytoplasmic hemoglobin solution in the voids between the convoluted spectrins to accommodate the shear deformation may also contribute to membrane shear. The extent of this contribution could be estimated by comparing determinations of $\bar{\eta_{\mathrm{mem}}}$ in hemoglobin-free red cells (ghosts) and in red cells. The available experimental data do not provide a clear-cut picture. Waugh [44] found a 36% lower value for $\bar{\eta_{\mathrm{mem}}}$ for ghosts compared to red cells. Nash and Meiselman [45], on the other hand, found a decrease of only 6%.By lowering a force-sensitive probe onto the ghost membrane, Heinrich et al. [1] probed the elastic thickness compressibility of the red cell membrane. The first elastic recoil was observed at a skeleton thickness of ~0.090 µm. An appreciable value was observed at ~0.050 µm. This value is in keeping with the results of an electron microscopic study [46]. The hardcore thickness was found by Heinrich et al. [1] at ~0.024 µm. Based on these numbers, the contribution of the envisaged mechanism could be estimated by applying the concept of flow through porous media.**1c** For the sake of completeness, the analogous mechanism with respect to the glycocalyx on the outside of the membrane is listed here.**1d** Consider a red cell after release from a static elongation—the standard experiment to determine $\bar{\eta_{\mathrm{mem}}}$. The contracting skeleton drags the Band 3 complexes through the bilayer as illustrated schematically in Figure 2. Figure 2A shows the smallest unit of the idealized skeleton in the relaxed state. The circles indicate the cross sections of the intrinsic portions of Band 3 complexes and Band 3 dimers drawn to scale as adopted from Burton and Bruce [47]. Drawn are 4 dimers. The average number is 3.5. The edge length of the triangle is 75 nm [48].Figure 2B shows two hexagons each composed of six triangles as in Figure 2A. For simplicity, the Band 3 complexes and free Band 3 dimers are not shown. The left hexagon is extended 1.5-fold. The right one is relaxed. The red arrows in the left hexagon indicate the direction in which the Band 3 complexes are dragged during the relaxation towards the positions shown in the right hexagon. Considering the motion of a single Band 3 complex, one is tempted to call the situation another type of skeleton slip. However as shown in the left hexagon, there is no net relative motion between the skeleton and bilayer. On average, the displacements of all Band 3 complexes cancel. This is a distinctive difference to skeleton slip which can be appreciated by comparing Figure 2B,C.Figure 2C sketches the situation during the aspiration of a membrane tongue into a small bore pipette. Here, the hexagon moves as a whole relative to the bilayer whereas in Figure 2B the hexagon just changes its shape and does not move. Besides the relative motion, the surface area changes as sketched in Figure 2C. This change is linked to the skeleton slip but is not quoted in the context of the distinctive difference.

### 3.2. Skeleton Slip

**2a** The friction due to the drag of the intrinsic proteins through the bilayer has been described in Section 2.2.**2b** For a contribution of cytoplasmic and ambient fluid flow through the layer of the skeletal proteins and the glycocalyx, the geometries of the respective experiments have to be considered. In TTing the shear stress effectively acts on the skeleton at the cytoplasmic side and on the glycocalyx on the outside. The 2D membrane flow would be slightly deformed due to the redistribution of the skeleton. However, the effect on energy dissipation is expected to be minute. During relaxation after release from an elongation, skeleton slip is directed from the attachment points towards the cell body. Again, the effect is expected to be small. The situation might be different during the aspiration of a membrane tongue. On the cytoplasmic side, the skeleton is squeezed like a sponge due to the increase in its local density. The same applies on the outside with the condensing glycocalyx. The last contribution could be estimated based on the observation of the fluid flow in the small gap between the membrane and the inner pipette wall [49].

### 3.3. Monolayer Slip

The two contributing mechanisms have been already described in Section 2.3 and are therefore just itemized here.
**3a** The flow of lipids around intrinsic proteins.**3b** The friction due to the sliding of the hydrocarbon chains past each other.

An effect of these mechanisms might be observable during TTing where the membrane flows over the rim of an elongated red cell. The curvature at the rim is higher than in the rest of the cell thus producing gradients in the surface density of both monolayers which lead to a steady state flow of lipids relative to the skeleton.

## 4. Suggested Steps to Widen the Knowledge of Membrane Mechanical Parameters

The following items are the ideas of an experimentalist. A realization will depend on the ratio of necessary effort and possible results. Steps 1 to 3 are targeted to evolve $\bar{\eta_{\mathrm{mem}}}$ from a descriptive parameter depending on the respective experiment towards a universal physical coefficient. Step 5 attempts to pinpoint the molecular mechanism of dynamic membrane shear. Steps 4 and 6 suggest the first steps to put numbers to the respective frictional coefficients.
The discordant behavior shown in Table 2 could be revisited by using the data in [18]. For each value of the TTF, the axis dimensions *L* and *W* have been determined and are available upon request. (The respective raw data have been sent to a number of groups but have never been used in conjunction.) Using these data, both μ and $\bar{\eta_{\mathrm{mem}}}$ could be fitted. The value of the bending stiffness (in slb) should be sufficient to prevent buckling during TTing. Since the strain hardening behavior of the shear elasticity of red cell membrane is not known, μ should depend on the 2D shear strain (ϵ) of the membrane. The deformation and TTF in steady state should allow the determination of μ(ϵ) and $\bar{\eta_{\mathrm{mem}}}$. The subsequent stop of the shear flow should allow to determine a second value for $\bar{\eta_{\mathrm{mem}}}$ from the recovery of the elongation. The double determination in the same model using the same data has not been done before.If the model uses the stress–strain law of Skalak et al. [50], the above calculations represent a test of a proposal of Dimitrakopoulos [51] who claimed that this law together with the small deformation value of μ measured by Lenormand et al. [52] describes various experiments sufficiently. Generally, the determined dependency μ(ϵ) indicates how close the constitutive equation of the respective model is to the real strain hardening behavior of the red cell membrane. The interpretation of the results requires some caution. At low red cell elongations, the membrane shear strains are small. As a consequence, the error in the fitted values of μ is larger the more the reference configuration of the model differs from reality. However, with increasing elongations, this error decreases.Most likely, the trend of decreasing $\bar{\eta_{\mathrm{mem}}}$ with increasing $\dot{\gamma }$ as observed previously will persist in step 1. As a refinement, the calculations could be repeated with a membrane viscosity $\eta_{\mathrm{mem}}$ depending on the 2D shear rate in the membrane.The variation in the raw data of *L* and *W* is rather large. After excluding extreme values, the range normalized by the mean value is about 0.3. To exclude the error due to this variation, step 2 could be performed for every single cell.In aspirating cell membrane into small bore pipettes, Chien et al. [8] found a second, much slower characteristic time ($\tau_{2}$) upon entry. The corresponding additional increase in tongue length was suggested to be due to skeleton slip [10]. If correct, it would allow a parametric fit of $c_{f}$. Since no raw data from these experiments are available, the following procedure is suggested. Use a model with an incompressible skeleton. Determine the time course of tongue length in a typical pipette experiment of [8]; e.g., pressure difference −18 mm $H_{2}O$, pipette radius 0.3 µm, μ=4.2 µN/m, and $\bar{\eta_{\mathrm{mem}}}=6\cdot 10^{-8}$ Pa s m. Add compressibility of the skeleton to the model and allow for skeleton slip. Repeat the calculation with $c_{f}=\left(\rho /\rho_{0}\right)144\cdot 10^{6}$ Ns/m³. If the tongue length increases further, adjust $c_{f}$ to obtain a value of $\tau_{2}=5$ s as found in [8]. This would constitute an experimental determination of $c_{f}$.If the tongue length decreases, the origin of the observed increase is not related to skeleton slip as suggested [10]. However, a close look at the recorded trace of the tongue length in Figure 7 of [8] shows a decreasing trend 20 s after the aspiration. This trend might be due to the modeled decrease. In order to be able to perform a parametric fit, a complete recording of the time course of the tongue length would be required.Another useful information in this respect might be the observation that the redistribution of the skeleton was complete after <1 min after aspiration of a membrane tongue into a pipette [49]. The time course was not given in this study probably because the required exposure time did not allow to record it. With 3τ=1 min, at least an upper bound for $c_{f}$ could be obtained.As noted in Section 3.1, the friction due to the drag of a single Band 3 complex or Band 3 dimer through the bilayer is the same in skeleton slip and dynamic membrane shear. Therefore, it should be possible to express $\bar{\eta_{\mathrm{mem}}}$ as a function of $c_{f}$ found in step 4. If this value corresponds to the values determined in step 1, mechanism 1d (Section 3.1) can be considered dominant. Otherwise, additional mechanisms have to be taken into account.As mentioned in Section 3.3, bcb may be local at the tips of strongly elongated and TTing red cells. To deduce a value of $c_{f}$ in monolayer slip, add slb with $2\cdot 10^{-19}$ J and bcb with $6\cdot 10^{-19}$ J (Considering each monolayer consisting of the same isotropic material, the ratio between the bending stiffnesses in bcb and slb is three. With a large variation, Waugh et al. [35] found a ratio of 3.4 in phospholipid vesicles.) as the respective stiffness parameters to the model in step 1. Choose the spontaneous curvature such that ratio of cell thickness in the dimple and the rim equals 0.55 [53,54]. Allow for monolayer slip during TTing and start with $c_{f}=0$. If the curvatures at low elongations and low TTFs compare well with observations but deviate at high elongations and high TTFs, increase the frictional coefficient until you get agreement for all shapes. This would constitute an experimental determination of $c_{f}$.

## 5. Consequences for the Modeling of Red Cell Experiments

A slow redistribution of skeleton density as found in [27] would have consequences for the reference configuration of the shear elasticity required in modeling. The circulation time through the vasculature is about 1 min. During a typical lap, the red cells are deformed in different shapes leaving not enough time to attain the steady state distribution of the skeleton in each of these shapes. In the long run, the distribution of the skeleton is therefore expected to be the mean of these (conceptual) steady state distributions weighted by their relative life time. This envisioned kind of average represents a modification of an equivalent suggestion that has been made earlier for the reference configuration in general [18,54,55].

The suggested average of skeleton density pertains to the situation in vivo. For parametric fits to experiments, the situation may be different. During handling of the blood after withdrawal as well as during the execution of an experiment, the envisioned average of skeleton density prevailing in the circulation may change. Based on the value of $c_{f}$ determined in Section 4 step 4, the time course of skeleton redistribution can be determined. Comparison of this time course to the history of the red cells in vitro could be used to choose/guess the appropriate reference configuration on which the modeling has to be based. The same argument applies to the relaxation of local bcb to global bcb based on the value of $c_{f}$ determined in Section 4 step 6.

## 6. Conclusions

Three contributions to the energy dissipation in the membrane under dynamic red cell deformations are identified. The first (dynamic membrane shear) corresponds to the well-known 2D membrane viscosity. Despite a lot of experimental and theoretical work, the underlying mechanisms on a molecular scale are not known and its frictional parameter ($\bar{\eta_{\mathrm{mem}}}$) remains descriptive; i.e., it depends on the respective experiment. For the second and third contributions (skeleton slip and monolayer slip), the underlying mechanisms on a molecular scale are known, however, the respective frictional coefficients are unknown. Steps to widen the knowledge are suggested. If successful, it is expected to determine the relative share of the three contributions to the energy dissipation in the various deformation modes human red cells are subjected to in vivo as well as in vitro. The values of the characteristic times associated with the second and third contributions will determine to what extent skeleton slip and monolayer slip actually contribute in a particular situation. Finally, it is expected that the improved modeling will facilitate the interpretation of experiments performed to provide information on the reference configuration of the red cell skeleton the last big unknown of mechanical membrane properties. 

## Figures and Tables

**Figure 1 biomolecules-13-00130-f001:**
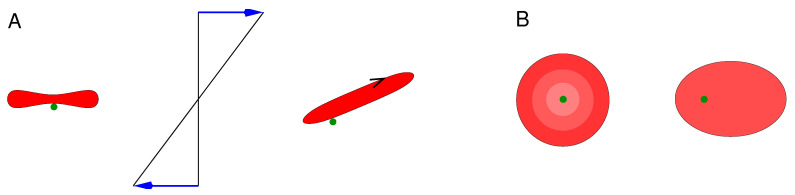
Schematic of the kinematics of the tank-tread (TT) motion. (**A**) View along the vorticity of a simple shear flow (middle). Left, a red cell at rest in edge-on orientation. Right, TTing under the imposed shear flow. The orientation of the elongated cell is almost stable due to a compensation of two turning moments [15]. The (black) arrow indicates the motion of the membrane. The dimple region is marked by a membrane-attached sphere (dark green). (**B**) View along the gradient of the simple shear flow. In an experiment, the TT motion is visualized by a back-and-forth motion of membrane-attached spheres [13].

**Figure 2 biomolecules-13-00130-f002:**
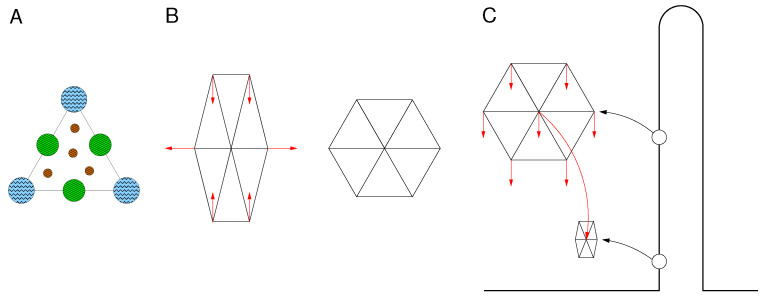
(**A**) Schematic of a triangle in the idealized network of the skeleton. The circles indicate the cross sections of the intrinsic portions of Band 3 complexes and Band 3 dimers drawn to scale. The diameters are 18.5, 15.8, 6.2 nm for the junction complex (blue), suspension complex (green), and Band 3 dimer (brown), respectively. In vivo, the six hetero dimers of spectrin connecting junction complexes and suspension complexes are strongly convoluted and rise above the bilayer plane [46]. Here, they are indicated by straight lines. (**B**) Schematic of two hexagons each composed of six triangles as in A. The left hexagon is elongated as in a static elongation at the start of a relaxation process. The right one is relaxed indicating the situation at the end of the relaxation process. The red arrows in the left hexagon indicate the direction of motion of the Band 3 complexes during the relaxation. (**C**) Schematic of the fate of a hexagon during the aspiration of a membrane tongue into a small bore pipette. The straight arrows indicate the direction of motion of the Band 3 complexes during the aspiration. The curved arrow indicates the displacement of the hexagon to its final position at static equilibrium.

**Table 1 biomolecules-13-00130-t001:** Conflicting results in $\bar{\eta_{\mathrm{mem}}}$ upon fitting the same experimental data.

$\eta_{0}$ (mPa s)	54	54	Axis Ratio of the	
$\dot{\gamma }$ (1/s)	8	130	Stress Free Shape	Reference
$\bar{\eta_{\mathrm{mem}}}\;(10^{-8}$ Pa s m)	24	4.5	0.9	Matteoli et al. [2]
$\bar{\eta_{\mathrm{mem}}}\;(10^{-8}$ Pa s m)	40	19.5	0.84	Tsubota [3]

**Table 2 biomolecules-13-00130-t002:** Comparison of the $\bar{\eta_{\mathrm{mem}}}$ values from different studies.

Fitted Parameter	Reference	$\bar{\eta_{\mathrm{mem}}}\;(10^{-8}$ Pa s m)
τ	Hochmuth et al. [11]	60–80
TTF	Tran-Son-Tay et al. [12]	10–8.6
τ	Guglietta et al. [4]	42
TTF	Matteoli et al. [2], Tsubota [3]	4.5–40

## Data Availability

Not applicable.

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
