# Peer review of "Energy Dissipation in the Human Red Cell Membrane"

_biomolecules, 2023, doi:10.3390/biom13010130_

Round 1

Reviewer 1 Report

The manuscript presents a comprehensive review of the current knowledge on the energy dissipation in human erythrocytes’ membranes. The author analyzes various contributions to this process including the in-plane shear deformation of membranes, the intermonolayer friction in the lipid matrix, and the relative motion of the membrane skeleton and the bilayer. The relevant mechanisms on molecular scale are summarized by providing experimental and in-silico estimates of the corresponding parameters such as the membrane shear viscosity, different frictional coefficients, the respective mechanical moduli of the bilayer, etc. The impact of the various characteristic times on the state of the erythrocyte membrane in circulation are analyzed by the author together with their influence on the adaptation of an in-silico model to the history of red cells in experiments.

For the sake of clarity, I would recommend the inclusion of a List of abbreviations summarizing all abbreviations with their definitions. The title “Suggested steps” of the fourth part could be extended to better present and reveal the author’s vision as well as to emphasize the objective of the presented item list.

Taking into account the quality of the analysis performed on the reviewed scientific data as well as the scientific soundness of the described perspectives I recommend the publication of the manuscript. I expect the results critically summarized in the presented review as well as the detailed perspectives presented therein to be of great interest to a broad readership of researchers in cellular and membrane biophysics and rheology.

Author Response

Most changes (blue) originate from the suggestions of the scientific editor. His requirement of a more structured text lead to objective changes as well.

The title has been changed to: Suggested steps to widen the knowledge of membrane mechanical parameters. A more specific list would unduly elongate the title. However as suggested by the scientific editor, at top of the section a short discussion is added of what the suggested steps will provide, if performed.

A section "Abbreviations" has been added at the end of the text. The items are grouped in textual abbreviations and symbols representing physical quantities. Each group is ordered alphabetically.

Reviewer 2 Report

The manuscript “Energy dissipation in the human red cell membrane” discusses the deformations that occur in red blood cells under the action of a shear flow. Author focuses on the specific type of red blood cell motion, called the tank treading motion and highlights the features of the interaction between the membrane and the cytoskeleton in this process. The relationship between the elastic parameters describing this motion is considered. In the final part of the manuscript, the author suggests possible ideas for future experiments describing tank treading motion. I believe that the manuscript can be published with some minor modifications.

1. In my opinion, the introduction to the manuscript needs to be expanded. Since Section 2.1 begins with estimates of various elastic parameters such as viscosity and shear modulus, it is necessary to briefly illustrate how these quantities affect the behavior of red blood cells in shear flow. In particular, one could mention various diseases associated with the loss of RBCs deformability.

2. In my opinion it would be proper to mention different types of cell movement in shear flow, such as flipping and rolling motion. Since the manuscript later concentrates on the tank treading motion, it would be plausible to indicate the conditions under which the tank treading/flipping transition occurs (e.g., see papers Abkarian, M., Faivre, M., & Viallat, A. (2007). Swinging of red blood cells under shear flow. Physical review letters, 98(18), 188302; Skotheim, J. M., & Secomb, T. W. (2007). Red blood cells and other nonspherical capsules in shear flow: oscillatory dynamics and the tank-treading-to-tumbling transition. Physical review letters, 98(7), 078301).

3. In my opinion the manuscript would greatly benefit from an explanatory drawing illustrating the tank treading motion.

4. Equation (2) implies the absence of transmission between monolayers, due to which the bilayer is deformed as a whole. At the same time, Helfrich's experiments considered membranes consisting of several bilayers that slip relative to each other by definition. Therefore, the bending modulus of a multilayer is the sum of the bending moduli of an individual bilayer, and there is no contradiction with formula (2).

Author Response

Most changes (blue) originate from the suggestions of the scientific editor. His requirement of a more structured text lead to objective changes as well.

1. Actually, I think membrane parameters such as viscosity or shear modulus influence the circulation only marginally. The only quantities that matter are the surface to volume ratio as in many anemias and the cytoplasmic viscosity as in sickle cell disease. Both quantities are not even mentioned in my perspective.

Rather, the perspective is directed to improve our knowledge in the basic science of the healthy red cell. As to the behavior in vitro, the perspective is directed to theoreticians who build computer models of the red cell. I don't have to bore these people with these basic things.

2. The modes of motion of more or less biconcave red cells below the threshold shear rate for TTing is a very interesting subject. Modeling these modes can serve to get information on the true reference configuration of the skeleton. However, a prerequisite to do this modeling is (at least in most cases) the knowledge of the influence of the energy dissipation. None of my suggested steps is directed to these complicated modes of motion. As I wrote in section 6, this would be the next big goal.

3. It would be easy for me to include a little film of a TTing red cell and to provide a schematic to explain the view in the rheoscope. However because of my own research, the perspective is biased towards TTing anyway. Therefore, I hesitate to increase this bias. Readers unfamiliar with TTing could find the respective information in the literature. After all, I did not provide this kind of explanatory information of the other experimental approaches mentioned in the perspective.

If the editor thinks that a film and a schematic should be added, I am prepared to do it.

4. Yes, you are right. Even if the bending stiffness measured by using the fluctuations of phospholipid vesicles was due to bcb alone, the bending stiffness of two bilayers would be twice the stiffness of a single bilayer. Although inconsequential, my thinking was wrong all these years - thank you. The changed text in section 2.3 is green.

Reviewer 3 Report

The manuscript by Thomas M. Fischer analyzes possible mechanisms of energy dissipation in the motion of red blood cells. The mechanisms are based on different scales of the motion: large-scale shear stress, intermediate-scale friction of the lipid bilayer and skeleton, and friction inside the bilayer. The manuscript is well organized and well written; its reading is interesting and enjoyable. I recommend it for publication with a minor corrections taking into account remarks listed below.

There are several inexactnesses in the text.

1) Page 1, middle: “The membrane consists of a phospholipid bilayer and a so-called membrane skeleton attached on the cytoplasmic side of the bilayer. The phospholipid bilayer is referred to as bilayer in the following.

            Strictly, the lipid bilayer of RBC plasma membrane comprises not only phospholipids, but other types of lipids as well (e.g., glycolipids). Thus, it is not phospholipid bilayer only, although phospholipids are the most abundant.

2) Page 1, middle: “It provides bending stiffness to the membrane and due to its essential incompressibility, it also conserves the surface area of the membrane.

            The membrane is not exactly incompressible. The lipid bilayer modulus of lateral compressibility is finite, about 250 mN/m [Rawicz, W., Olbrich, K. C., McIntosh, T., Needham, D., & Evans, E. (2000) Effect of chain length and unsaturation on elasticity of lipid bilayers Biophysical journal, 79(1), 328-339]. A membrane typically withstand about 4% of lateral area increase. Thus, the surface area of the membrane is not exactly constant.

3) Page 1, bottom: “In silico models typically treat the membrane as a 2D sheet with no thickness.

            “In silico” usually refers to molecular dynamics-related methods, i.e., numerical solution of Newtonian motion equations for each atom (or group of atoms) in a 3D simulation box. In such approaches the membrane is considered as essentially three-dimensional structure; the motion and force interactions of each atom of the membrane (and water) are taken into account. Typically, continuum theories and models treat the membrane as a 2D sheet with no thickness, but, as a rule, they are not called “in silico”, as far as I know.

4) Page 5, middle: “Although the binary mixture indicated a value for the frictional coefficient about 10 times larger than the pure phospholipid, a single order of magnitude was given for the frictional coefficient (108 Ns/m3).

            From the phrase it follows that the frictional coefficients are simultaneously close to each other and 10 times different. Which statement is correct?

5) Page 5, bottom: “The extent of this contribution could estimated …

            Probably, “be” is missing in between “could” and “estimated”.

6) Page 7, bottom: “… producing gradients in monolayer density …

            Strictly, the monolayer density is constant if the monolayer is considered as a 3D body, as the bulk modulus of a membrane is extremely high (~1010 J/m3). The surface density of a monolayer can be non-constant.

7) Page 7, Legend of Figure 1.

            The abbreviations “JCs and SCs” were not defined.

8) Page 7, bottom: “As a simple 2D geometry, two flat portions of membrane connected by one half of a circular cylinder are considered.

            “A circular cylinder” is an incorrect term – in geometry the cylindrical surface is composed of straight lines, by definition. “A circular cylinder” is a toroidal surface.

Author Response

Most changes (blue) originate from the suggestions of the scientific editor. His requirement of a more structured text lead to objective changes as well.

1. Phospholipid is replaced by lipid in the context of the bilayer of red cells.

2. The biggest isotropic stress of all situations considered in the perspective is the experiment of Discher Mohandas and Evans 1994 in Science where the red cells were aspirated at a pressure difference 0.02 atm to obtain a spherical shape outside of the pipette. An estimate of the upper bound of the resulting isotropic membrane tension results in a value 1.4 dyn/cm which is almost an order of magnitude below the value for red cell lysis (Evans Waugh and Melnik 1976 Biophsical J.). Using your number of 4%, this would result in an increase of about 0.4%. In the other situations considered, the increase can be expected to much be lower. In this spirit, the word "essential" is a correct description. To express the above limitation the objected sentence has been split in three (red in section 1).

3. “In silico” has been eradicated without colorizing the changes.

4. Even if we knew why Evan Evans left the reader with this puzzling information, there is no need to transfer this uncertainty to the reader of this perspective. The puzzling half sentence was deleted.

5. Thank you, but working on the suggestions of the scientific editor, I noticed the mistake already myself.

6. The sloppy wording has been corrected (section 3.3 red)

7. The legend has been corrected.

8. In all links I checked, it says right circular cylinder. Here are 4 examples:
https://www.embibe.com/exams/cylinder/
https://www.cuemath.com/geometry/cylinder/
https://www.merriam-webster.com/dictionary/cylinder
https://mathmonks.com/cylinder

Therefore, circular cylinder was kept in section 2.2.

As to the occurrence in section 3.3, I noticed an inexactness in my analysis and cancelled the whole paragraph.